# Learning Object-Centric Neural Scattering Functions for Free-viewpoint Relighting and Scene Composition

**Hong-Xing Yu**[*1], **Michelle Guo**[*1], **Alireza Fathi**[2], **Yen-Yu Chang**[1], **Eric Ryan Chan**[1], **Ruohan Gao**[1], **Thomas Funkhouser**[2], **Jiajun Wu**[1]

[1] *Stanford University*     [2] *Google Research*     [*] *Contributed equally.*

**Reviewed on OpenReview:** *https://openreview.net/forum?id=NrfSRtTpN5*

## Abstract

Photorealistic object appearance modeling from 2D images is a constant topic in vision and graphics. While neural implicit methods (such as Neural Radiance Fields) have shown high-fidelity view synthesis results, they cannot relight the captured objects. More recent neural inverse rendering approaches have enabled object relighting, but they represent surface properties as simple BRDFs, and therefore cannot handle translucent objects. We propose Object-Centric Neural Scattering Functions (OSFs) for learning to reconstruct object appearance from only images. OSFs not only support free-viewpoint object relighting, but also can model both opaque and translucent objects. While accurately modeling subsurface light transport for translucent objects can be highly complex and even intractable for neural methods, OSFs learn to approximate the radiance transfer from a distant light to an outgoing direction at any spatial location. This approximation avoids explicitly modeling complex subsurface scattering, making learning a neural implicit model tractable. Experiments on real and synthetic data show that OSFs accurately reconstruct appearances for both opaque and translucent objects, allowing faithful free-viewpoint relighting as well as scene composition. Project website with video results: `https://kovenyu.com/OSF`.

## 1 Introduction

Modeling the geometry and appearance of 3D objects from captured 2D images is central to many applications in computer vision, graphics, and robotics, such as shape reconstruction (Mescheder et al., 2019; Park et al., 2019; Chang et al., 2015), view synthesis (Hedman et al., 2018; Mildenhall et al., 2020; Sitzmann et al., 2019a), relighting (Zhang et al., 2021c; Bi et al., 2020b;a), and object manipulation (Simeonov et al., 2021). Traditional inverse rendering approaches (Zhou et al., 2013; Nam et al., 2018) focus on directly estimating objects' shape and material properties; they then compose them using the estimated surface properties. These explicitly reconstructed object representations often lead to visible artifacts or limited fidelity when rendering images using the recovered object assets.

Recently, neural implicit methods have attracted much attention in image-based appearance modeling. They implicitly represent objects and scenes using deep neural networks (Lombardi et al., 2019; Sitzmann et al., 2019a;b). Neural Radiance Fields (NeRFs) (Mildenhall et al., 2020) are one of the most representative methods due to their high-fidelity view synthesis results. NeRFs implicitly model a scene's volumetric density and radiance via coordinate-based deep neural networks, which can be learned from images via direct volume rendering techniques (Max, 1995). However, NeRFs only reconstruct the outgoing radiance fields under a fixed lighting condition. Thus, they cannot relight the captured objects or compose multiple objects into new scenes.

To address this problem, a few neural inverse rendering methods (Zhang et al., 2021b;a; Boss et al., 2021a; Zhang et al., 2021c) have been proposed to jointly estimate lighting, material properties, and geometry from

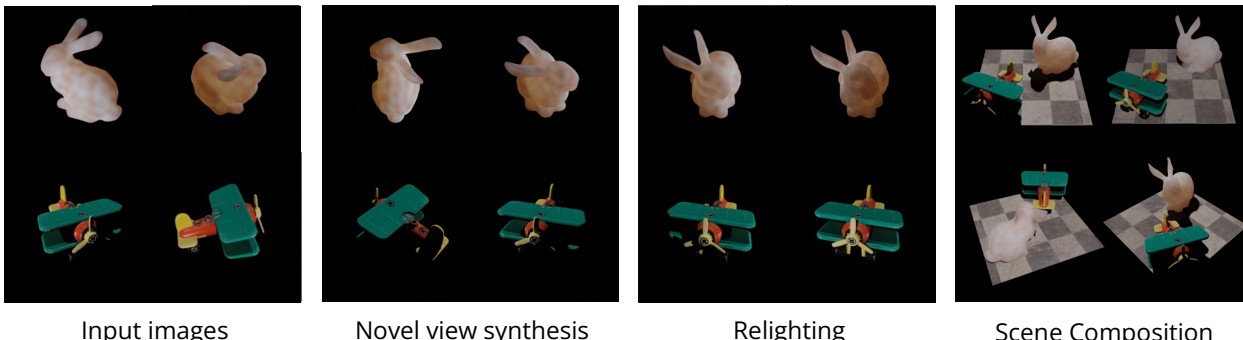

| Input images | Novel view synthesis | Relighting | Scene Composition |

Figure 1: We propose a new neural object representation, Object-Centric Neural Scattering Functions (OSFs), to reconstruct object appearance from only images. OSFs can handle objects with complex materials or shapes (e.g., the translucent bunny), and support both free-viewpoint relighting and scene composition.

images. The disentanglement of illumination enables relightable object appearance modeling. However, these methods all assume simple Bidirectional Reflectance Distribution Functions (BRDFs) when modeling surface properties, without taking subsurface light transport into consideration[*]. Therefore, they can only model the appearance of opaque objects and cannot handle translucent objects with complex material properties.

We observe that subsurface scattering effects are the key to relighting translucent objects. However, accurately modeling subsurface scattering can be complex and even intractable for neural implicit representations, because each sampling operation for numerical integration requires a complete forward pass of the network. This leads to a prohibitive computational cost in both time and memory.

To address this challenge, we propose Object-Centric Neural Scattering Functions (OSFs) for image-based relightable neural appearance reconstruction. An OSF learns to approximate the *cumulative radiance transfer function*, which models radiance transfer from a distant light to any outgoing direction at any spatial location for an object. This enables the use of a volumetric rendering formulation to learn from images, while being able to approximate both opaque and translucent object appearances. Thus, OSFs allow free-viewpoint relighting from only 2D images (see Figure 1).

Modeling object-level light transport allows neural scene composition with trained OSF models, as light transport effects such as object shadows and indirect lighting can be naturally incorporated. To demonstrate this, we present a simple and computationally tractable method for scene composition. To further accelerate neural scene rendering, we introduce two improvements on OSFs scene rendering that allow magnitudes of acceleration without compromising rendering quality.

Experiments on both real and synthetic objects demonstrate the effectiveness of our OSF formulation. We show that OSFs enable free-viewpoint relighting of both translucent and opaque objects by learning from only 2D images. We also show that in scene composition using the reconstructed objects, our approach significantly outperforms baseline methods.

In summary, our contributions are threefold. First, we introduce the concept of the cumulative radiance transfer function that models object-centric light transport. Second, we propose the Object-Centric Neural Scattering Functions (OSFs) that learn to approximate the cumulative radiance transfer function from only 2D images. OSFs allow free-viewpoint relighting and scene composition of *both opaque and translucent objects*. Third, we show experimental results on both real and synthetic objects, demonstrating the effectiveness of OSFs in free-viewpoint relighting and scene composition.

## 2 Related Work

**Neural appearance reconstruction.** Traditional methods use Structure-From-Motion (Hartley & Zisserman, 2003) and bundle adjustment (Triggs et al., 1999) to reconstruct colored point clouds. More recently, a number of learning-based novel view synthesis methods have been proposed, but they require 3D geometry

---

[*]We use "simple BRDFs" to refer to BRDFs that do not consider subsurface scattering.

as inputs (Hedman et al., 2018; Thies et al., 2019; Meshry et al., 2019; Aliev et al., 2020; Martin-Brualla et al., 2018). Recently, neural volume rendering approaches have been used to reconstruct scenes from only images (Lombardi et al., 2019; Sitzmann et al., 2019a). However, the rendering resolution of these methods is limited by the time and computational complexity of the discretely sampled volumes. To address this issue, Neural Radiance Fields (NeRFs) (Mildenhall et al., 2020) directly optimizes a *continuous* radiance field representation using a multi-layer perceptron. This allows for synthesizing novel views of realistic images at an unprecedented level of fidelity. Various extensions of NeRF have also been proposed to improve efficiency (Liu et al., 2020; Yu et al., 2021; Garbin et al., 2021; Reiser et al., 2021), generalization (Niemeyer & Geiger, 2021; Yu et al., 2022; Yang et al., 2021), quality (Oechsle et al., 2021; Barron et al., 2021a;b; Verbin et al., 2021; Zhang et al., 2020), compositionality (Ost et al., 2021) etc. While these neural methods produce high-quality novel views of a scene, they do not support relighting. In contrast, our approach aims at relightable appearance reconstruction.

**Learning-based relighting.** Learning-based methods for relighting without explicit geometry have been proposed (Sun et al., 2019; Xu et al., 2018; Zhou et al., 2019), but they are not 3D-aware. Following the recent surge of neural 3D scene representations and neural differentiable rendering (Tewari et al., 2020; Mildenhall et al., 2020), neural relightable representations (Bi et al., 2020b;a; Boss et al., 2021a; Srinivasan et al., 2021; Zhang et al., 2021c; Baatz et al., 2021) and neural inverse rendering methods (Zhang et al., 2021b;a; Boss et al., 2021b) have been developed to address this limitation. However, most of them assume simple BRDFs which do not consider subsurface scattering in their formulations, and thus they cannot model translucent object appearances for relighting. In contrast, our method allows the reconstruction of both opaque and translucent object appearances. Some existing methods also approximate subsurface light transport for participating media (Kallweit et al., 2017; Zheng et al., 2021). While Kallweit et al. (2017) demonstrate high-quality scattering, they require 3D groundtruth data whereas our method learns from only images. Zheng et al. (2021) focus on relevant tasks as we do, using specific designs such as spherical harmonics for scattering and visibility modeling. In contrast, our model is simple and allows adopting recent advances in accelerating neural rendering. We demonstrate interactive frame rate in rendering by our KiloOSF.

**Precomputed light transport.** Our OSFs learn to approximate a cumulative radiance transfer function, which is based on the classical idea from the precomputed light transport methods including the precomputed radiance transfer (Sloan et al., 2002) (also see (Lehtinen, 2007) and (Ramamoorthi, 2009) for overviews). These methods map incoming basis lighting to outgoing radiance, while we additionally learn to encapsulate the object-specific light transport at the visible surface. Related ideas include the classical reflectance field (Debevec et al., 2000; Garg et al., 2006). Our method is also related to classical techniques for modeling aggregate scattering behavior (Moon et al., 2007; Lee & O'Sullivan, 2007; Meng et al., 2015; Blumer et al., 2016). These techniques consider modeling the full asset-level internal scattering, a more general form than our cumulative radiance transfer. Our cumulative radiance transfer function borrows the general idea from these classical methods and adapts to neural implicit volumetric rendering framework for relightable and compositional appearance modeling from images. This adaptation allows using the power of deep networks to approximate complex light transport. Recently a few neural methods have been proposed to model light transport of complex materials (Kuznetsov, 2021; Baatz et al., 2021), e.g., Baatz et al. (2021) represent surface texture as neural reflectance fields which is technically relevant to ours. However, they need meshes as explicit object geometry, while ours learns object representations from only images.

## 3 Approach

We aim for relightable neural appearance reconstruction of both opaque and translucent objects from 2D images. To achieve this, we propose Object-centric neural Scattering Functions (OSFs) that learn to approximate subsurface scattering effects in a computationally tractable way. In the following, we first introduce our OSF formulation and discuss its relations to phase functions and BSSRDFs (Sec. 3.1). Then we show how we can model translucent objects (Sec. 3.2) and opaque objects (Sec. 3.3) with OSFs. Finally, we discuss how to learn OSFs from 2D images (Sec. 3.4) and compose multiple learned OSFs (Sec. 3.6).

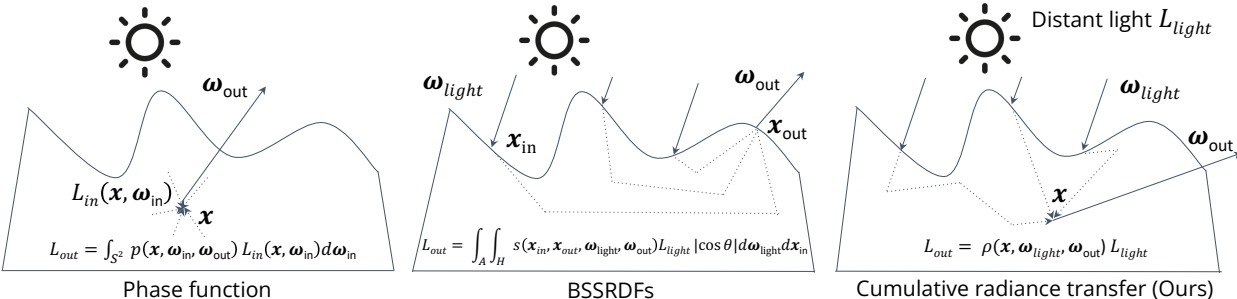

Figure 2: **Relations to classical graphics concepts for modeling subsurface scattering effects.** The process of light transport from an unobstructed distant light to a point $\mathbf{x}$ inside a translucent object is complex: 1) the light (radiance) hit a point on the visible surface area; 2) the surface transmits some amount of the light into the object interior; 3) the transmitted light scatters within the object; 4) some light finally arrives at $\mathbf{x}$ from incoming directions $\boldsymbol{\omega}_{\text{in}}$; 5) the arrived light scatters to directions $\boldsymbol{\omega}_{\text{out}}$; and 6) this process happens at all points on the visible surface, and the final outcome is the *accumulated* result. Phase functions only model 4) and 5), so one has to account for the other steps by multiple integrations. BSSRDFs encapsulate steps 2) - 5), so they require two integrations for step 1) and step 6). In contrast, our OSF formulation encapsulates the entire process that takes all steps into account. This abstraction of modeling the whole process as cumulative radiance transfer eliminates the need of any integration, making it computationally tractable to use in neural volume rendering.

### 3.1  OSF Formulation

Our OSFs are based on Neural Radiance Fields (NeRFs) (Mildenhall et al., 2020), which learn view-dependent emitted radiance at any point in space. NeRFs can be formulated by $\text{NN}_\Theta : (\mathbf{x}, \boldsymbol{\omega}_{\text{out}}) \to (L_{\text{out}}, \sigma)$, where NN denotes a learnable deep neural network with parameters $\Theta$, $\mathbf{x}$ denotes a 3D spatial location, $\boldsymbol{\omega}_{\text{out}}$ denotes an outgoing radiance direction, $L_{\text{out}}$ denotes the outgoing radiance, and $\sigma$ denotes the volumetric density. Inspired from the principles of volume rendering (Kajiya & Von Herzen, 1984), the computational model for the radiance $L(\mathbf{o}, \boldsymbol{\omega}_{\text{out}})$ arriving at the camera location $\mathbf{o}$ from direction $\boldsymbol{\omega}_{\text{out}}$ (a.k.a. the radiance of a camera ray $\mathbf{r}(t) = \mathbf{o} - t\boldsymbol{\omega}_{\text{out}}$) with near and far bounds $t_n$ and $t_f$ is given by:

$$L(\mathbf{o}, \boldsymbol{\omega}_{\text{out}}) = \int_{t_n}^{t_f} T(t)\sigma(\mathbf{r}(t))L_{\text{out}}(\mathbf{r}(t), \boldsymbol{\omega}_{\text{out}})dt, \tag{1}$$

where $T(t) = \exp(-\int_{t_n}^{t} \sigma(\mathbf{r}(s))ds)$ denotes the transmittance along the ray from $t_n$ to $t$.

While a NeRF can render high-fidelity novel views of a captured object, it does not support relighting because it is based on the absorption-plus-emission model (Max, 1995), which itself is light source–agnostic. Namely, NeRFs model $L_{\text{out}}$ as the *emitted* radiance after all light transport has occurred. To address this limitation, we propose OSFs that model $L_{\text{out}}$ as the *scattered* radiance due to external light sources instead. More specifically, OSFs learn to approximate the cumulative radiance transfer from an unobstructed distant light, in addition to the volume density:

$$\text{NN}_\Theta : (\mathbf{x}, \boldsymbol{\omega}_{\text{light}}, \boldsymbol{\omega}_{\text{out}}) \to (\rho, \sigma), \tag{2}$$

where $\rho(\mathbf{x}, \boldsymbol{\omega}_{\text{light}}, \boldsymbol{\omega}_{\text{out}})$ denotes the *cumulative radiance transfer function*, which is defined as the ratio of two quantities. The numerator is the outgoing radiance $L_{\text{out}}(\mathbf{x}, \boldsymbol{\omega}_{\text{out}})$ in the direction of $\boldsymbol{\omega}_{\text{out}}$ at location $\mathbf{x}$ that is caused by an unobstructed distant light $L_{\text{light}}(\boldsymbol{\omega}_{\text{light}})$ from direction $\boldsymbol{\omega}_{\text{light}}$. The denominator is $L_{\text{light}}(\boldsymbol{\omega}_{\text{light}})$. It is *cumulative* about all paths that $L_{\text{light}}(\boldsymbol{\omega}_{\text{light}})$ scatters to $\mathbf{x}$. See Figure 2 for an illustration.

Given the cumulative radiance transfer function $\rho$, the scattered outgoing radiance $L_{\text{out}}$ is given by

$$L_{\text{out}}(\mathbf{x}, \boldsymbol{\omega}_{\text{out}}) = \int_{S^2} \rho(\mathbf{x}, \boldsymbol{\omega}_{\text{light}}, \boldsymbol{\omega}_{\text{out}})L_{\text{light}}(\boldsymbol{\omega}_{\text{light}})d\boldsymbol{\omega}_{\text{light}}. \tag{3}$$

We assume the object itself does not emit light. We argue that this distant light assumption in $\rho$ is acceptable and can allow relighting with environment maps that are also distant light.

**Phase Functions vs. BSSRDFs. vs. Cumulative Radiance Transfer.** The cumulative radiance transfer function $\rho$ approximates a complex radiance transfer process from an unobstructed distant light to an outgoing direction at a location, including those inside the object. It is conceptually akin to a phase function that describes the scattered radiance distribution at a specific location in a scattering process. It is also related to the Bidirectional Scattering Surface Reflectance Distribution Functions (BSSRDFs) (Jensen et al., 2001). See Figure 2 for a detailed comparison of these concepts.

Below we list the major distinctions against phase functions:

- Generally, a phase function is local, i.e., it depends only on the material/media (and in some cases also geometry) at the location. In contrast, the cumulative radiance transfer function $\rho$ is object-specific and spatially varying even if the object is made from a uniform homogeneous material. This is because $\rho$ potentially needs to account for a complex process, including the light transport for all surface points that are visible to the distant light, the subsurface scattering effects, and the radiance transfer at a specific location. Thus, this process depends on both the material and the object geometry at $\mathbf{x}$, as well as the light direction.

- Phase functions for some naturally occurring isotropic media only need 1 Degree of Freedom (DoF) for angular distribution, i.e., the cosine between incoming and outgoing radiance directions (see Chapter 11 in Pharr et al. (2016)). However, $\rho$ generally needs 4 DoF for angular distribution: 2 DoF of light direction because the light direction can interact with object geometry to affect the incoming radiance distribution at any given $\mathbf{x}$, and 2 DoF of outgoing direction as we want to model view-dependent effects as in NeRFs (Mildenhall et al., 2020).

Compared to BSSRDFs which are surface models, $\rho$ is used in a volumetric rendering process. OSFs do not explicitly model surface properties.

## 3.2 Modeling Translucent Objects with OSFs

Our OSF formulation can model the appearance of translucent objects in a computationally tractable way for volumetric neural implicit methods, while directly modeling the radiance transfer in a volumetric scattering process is intractable. In direct modeling, $L_{\text{out}}(\mathbf{x}, \boldsymbol{\omega}_{\text{out}}) = \int_{S^2} p(\mathbf{x}, \boldsymbol{\omega}, \boldsymbol{\omega}_{\text{out}}) L_{\text{in}}(\mathbf{x}, \boldsymbol{\omega}) d\boldsymbol{\omega}$, where $p$ denotes a phase function, $L_{\text{in}}$ denotes the incoming radiance at location $\mathbf{x}$ from the direction $\boldsymbol{\omega}$. The recursive dependency of $L_{\text{in}}$ over the unit sphere in the media makes the equation very difficult to solve (Novák et al., 2018; Max & Chen, 2005). From the computational perspective, to numerically compute this integral to train the neural network, one needs to account for the whole sphere (due to subsurface scattering) through dense sampling. However, for neural implicit methods, each sample requires a forward pass of the deep network, leading to a high computation cost. Moreover, training deep networks often requires saving intermediate variables for auto-differentiation (Paszke et al., 2017), which demands a massive amount of GPU memory for dense sampling. As a result, it can become computationally intractable to train the neural network in this way. In contrast, during training, we use lighting setups that match well the assumption of a single directional light source; the spherical integration in Eq. (3) thus reduces to a single evaluation. The key feature of OSFs is that they approximate the complex cumulative radiance transfer process using the expressive power of deep neural networks.

## 3.3 Modeling Opaque Objects with OSFs

Apart from modeling translucent objects, OSF formulation is also able to model opaque objects. Theoretically, the formulation may subsume surface light transport for opaque convex objects. To see this, consider a single distant light source from $\boldsymbol{\omega}_{\text{light}}$, the light transport at a visible surface point can be described by

$$L_{\text{out}}(\mathbf{x}_{\text{surf}}, \boldsymbol{\omega}_{\text{out}}) = f(\mathbf{x}_{\text{surf}}, \boldsymbol{\omega}_{\text{light}}, \boldsymbol{\omega}_{\text{out}}) L_{\text{light}} \left| \mathbf{n}(\mathbf{x}_{\text{surf}}) \cdot \boldsymbol{\omega}_{\text{light}} \right|, \tag{4}$$

where $f$ denotes a BRDF, $\mathbf{x}_{\text{surf}}$ denotes the first surface point hit by the camera ray $\mathbf{r}$, and $\mathbf{n}(\mathbf{x})$ denotes the unit normal vector at $\mathbf{x}$. We have used the single distant light assumption to solve the integration. Now we consider an ideal case, where an OSF learns that $\sigma(\mathbf{x}) = \delta(\mathbf{x} - \mathbf{x}_{\text{surf}})$ for all $\mathbf{x}_{\text{surf}}$. This solves the integration in Eq. (1) and gives

$$L(\mathbf{o}, \boldsymbol{\omega}_{\text{out}}) = L_{\text{out}}(\mathbf{x}_{\text{surf}}, \boldsymbol{\omega}_{\text{out}}) = \rho(\mathbf{x}_{\text{surf}}, \boldsymbol{\omega}_{\text{light}}, \boldsymbol{\omega}_{\text{out}}) L_{\text{light}}. \tag{5}$$

We can see that if the learned OSF satisfies $\rho(\mathbf{x}_{\text{surf}}, \boldsymbol{\omega}_{\text{light}}, \boldsymbol{\omega}_{\text{out}}) = f(\mathbf{x}_{\text{surf}}, \boldsymbol{\omega}_{\text{light}}, \boldsymbol{\omega}_{\text{out}}) |\mathbf{n}(\mathbf{x}_{\text{surf}}) \cdot \boldsymbol{\omega}_{\text{light}}|$ for all surface points, then it subsumes the surface light transport with the BRDF. Although a neural network cannot perfectly represent a delta function for $\sigma(\mathbf{x})$, in our experiments, we observe that empirically OSFs learn to approximate the surface light transport for opaque objects of both convex and concave shapes.

## 3.4 Learning OSFs from Images

We aim to learn OSFs from only 2D images, allowing easy capture of real objects. As discussed in Section 3.1, we prefer a tractable training setup. Thus, we propose capturing object images using only a single distant light source, which leads to a light source radiance function $L_{\text{light}}(\boldsymbol{\omega}) = L_0 \delta(\boldsymbol{\omega} - \boldsymbol{\omega}_0)$, where $L_0$ and $\boldsymbol{\omega}_0$ are obtained from the capture setup. We can define the sampling Probabilistic Density Function (PDF) for lighting as $\text{PDF}_{\text{light}}(\boldsymbol{\omega}) = \delta(\boldsymbol{\omega} - \boldsymbol{\omega}_0)$ and this gives an analytic solution of the integration in Eq. (3):

$$L_{\text{out}}(\mathbf{x}, \boldsymbol{\omega}_{\text{out}}) = \mathop{\mathbb{E}}_{\boldsymbol{\omega} \sim \text{PDF}_{\text{light}}(\boldsymbol{\omega})} \left[ \rho(\mathbf{x}, \boldsymbol{\omega}, \boldsymbol{\omega}_{\text{out}}) \frac{L_{\text{light}}(\boldsymbol{\omega})}{\text{PDF}_{\text{light}}(\boldsymbol{\omega})} \right] \tag{6}$$

$$= \rho(\mathbf{x}, \boldsymbol{\omega}_0, \boldsymbol{\omega}_{\text{out}}) L_0. \tag{7}$$

We capture images of the object-to-reconstruct under different $\omega_0$ in a dark room. We describe our accessible real data capture setup using two iPhones in Section 4.1.

Given Eq. (7), the camera ray radiance $L(\mathbf{o}, \boldsymbol{\omega}_{\text{out}})$ defined in Eq. (1) is trivially differentiable with respect to OSFs. We follow the learning strategy from NeRFs (Mildenhall et al., 2020), which models the pixel color of a ray in an analog to radiance instead of the raw radiometric quantity to simplify learning. This may be interpreted as allowing the neural network to implicitly deal with the camera response that maps camera ray radiance to pixel color (also see a recent discussion in Mildenhall et al. (2021)). We also follow NeRFs to use quadrature to compute Eq. (1), and use positional encoding and hierarchical volume sampling to facilitate training (i.e., we use a coarse network for coarse sampling a ray to give a rough estimation of transmittance distribution, and then use a fine network to do additional informed sampling to further reduce variance). The loss function for learning an OSF is thus defined as

$$\mathcal{L}(\Theta) = \mathbb{E}_{\mathbf{r}} \left[ \|L_{\text{coarse}}(\mathbf{o}, \boldsymbol{\omega}_{\text{out}}) - C(\mathbf{r})\|^2 + \|L_{\text{fine}}(\mathbf{o}, \boldsymbol{\omega}_{\text{out}}) - C(\mathbf{r})\|^2 \right], \tag{8}$$

where $C(\mathbf{r})$ denotes the groundtruth pixel color (which is up to a constant of the radiance assuming a linear camera response) for ray $\mathbf{r} = \mathbf{o} - t\boldsymbol{\omega}_{\text{out}}$, $L_{\text{coarse}}(\mathbf{o}, \boldsymbol{\omega}_{\text{out}})$ denotes the predicted radiance by the coarse network, and $L_{\text{fine}}(\mathbf{o}, \boldsymbol{\omega}_{\text{out}})$ denotes the predicted radiance by the fine network. During testing, we only use the predicted radiance of the fine network.

## 3.5 KiloOSF for Accelerating Rendering

Neural volumetric rendering is slow. We show a complexity analysis in the following subsection. To accelerate OSFs, in the same spirit as KiloNeRF (Reiser et al., 2021), we introduce a variant of our model called KiloOSF that represents the scene with a large number of independent and small MLPs. KiloOSF represents each object with thousands of small MLPs, each responsible for a small portion of the object, making each individual MLP sufficient for high-quality rendering.

Specifically, we subdivide each *object* into a uniform grid of resolution $\mathbf{s} = (s_x, s_y, s_z)$. Each grid cell is of 3D index $\mathbf{i} = (i_x, i_y, i_z)$. We define a mapping $m$ from position $\mathbf{x}$ to index $\mathbf{i}$ through spatial binning as follows:

$$m(\mathbf{x}) = \lfloor (\mathbf{x} - \mathbf{b}_{\text{min}})/(\mathbf{b}_{\text{max}} - \mathbf{b}_{\text{min}}) \rfloor, \tag{9}$$

where $\mathbf{b}_{\min}$ and $\mathbf{b}_{\max}$ are the respective minimum and maximum bounds of the axis-aligned bounding box (AABB) enclosing the object. For each grid cell, a small MLP neural network with parameters $\Theta(\mathbf{i})$ is used to represent the corresponding portion of the object. Then, the radiance transfer and density values at a point $\mathbf{x}$ can be obtained by first determining the index $m(\mathbf{x})$ responsible for the grid cell that contains this point, then querying the respective small MLP:

$$\mathrm{NN}_{\Theta(m(\mathbf{x}))} : (\mathbf{x}, \boldsymbol{\omega}_{\mathrm{light}}, \boldsymbol{\omega}_{\mathrm{out}}) \to (\rho, \sigma), \tag{10}$$

Following KiloNeRF, we use a "training with distillation" strategy. We first train an ordinary OSF model for each object and then distill the knowledge of the teacher model into the KiloOSF model. We also use empty space skipping and early ray termination to increase rendering efficiency (Reiser et al., 2021).

### 3.6 Composing Multiple Learned OSFs

We show how we use learned OSFs for visually plausible scene composition. Due to the complexity of the problem and the approximations made by OSFs, physical correctness may not be guaranteed in the following discussion. Below we present a simple and computationally tractable approximation for composing multiple learned OSFs. We leave the design of physically correct scene composition algorithms as future work.

We restrict our discussion to a scene with multiple objects represented by OSFs, and we assume a single distant light source to illuminate the scene. In this case, the light distribution in Eq. (3) can potentially contain two parts: one for direct lighting, and the other for indirect lighting. We discuss them separately below, and then provide an algorithm for rendering a composed scene.

**Direct lighting.** Direct lighting is assumed to be a distant light, so the major point to consider is solving visibility. Notice that although OSFs allow reconstructing translucent object appearances, we do not model advanced light transport effects including transmitted lights or caustics in this work, but only consider shadows caused by light visibility. While solving for hard visibility can be tricky for OSFs, because they do not have the concept of surface just like NeRFs, the transmittance $T$ in Eq. (1) can be seen as a soft measure for visibility. Therefore, we use a shadow ray $\mathbf{r}_{\mathrm{shadow}}(t) = \mathbf{x} - t\boldsymbol{\omega}_{\mathrm{light}}$ to solve for visibility at $\mathbf{x}$. We volume-render this shadow ray for all other OSFs in the scene with compositional rendering (Niemeyer & Geiger, 2021), and use the obtained transmittance $T(t) = \exp(-\int_{t_n}^{t} \sigma(\mathbf{r}_{\mathrm{shadow}}(s))ds)$ to serve as visibility: $L_{\mathrm{direct}} = T(t_f)L_0$, where we abuse $t_f$ to denote the far bound of the shadow ray and $L_0$ to denote the light radiance. Given $L_{\mathrm{direct}}$, we compute outgoing radiance using Eq. (7).

Notice that this is a biased estimation for translucent objects, because the visibility is solved by the ray cast from a single point $\mathbf{x}$, while the cumulative transfer function $\rho$ assumes the full light visibility for the object surface (Figure 2). Therefore, if a translucent object is partially occluded from the distant light, the outgoing radiance is underestimated for those points casting occluded shadow rays, and overestimated for those points casting unobstructed shadow rays. This also happens for concave opaque objects due to global transport. But for ideally learned OSFs of convex opaque objects (i.e., they degenerate to BRDFs) as discussed in section 3.3, the estimation is unbiased.

**Indirect lighting.** Besides direct lighting, we also want to account for indirect lighting in evaluating Eq. (3). To do this, we use Monte Carlo integration with uniform sampling of solid angles. Specifically, we cast a ray from $\mathbf{x}$ to the sampled direction, and volume-render the ray using Eq. (1) to obtain the incoming radiance. Due to high computational cost, we only consider one-bounce indirect lighting. Similar to direct lighting, the estimation is biased for translucent objects, because $\rho$ assumes distant light, while inter-reflection from objects do not meet this assumption. Nonetheless, our experiments show that our approximation can create reasonably faithful scene composition results in Sec. 4.3.

**Excluding non-intersecting rays to accelerating scene composition.** The standard rendering procedure must be repeated per pixel to render an image. The total cost of rendering a single image with $N_{\mathrm{pixel}}$ pixels and $N_{\mathrm{object}}$ objects is thus

$$\mathcal{O}((N_{\mathrm{pixel}}N_{\mathrm{sample}}N_{\mathrm{object}})(N_{\mathrm{light}}N_{\mathrm{sample}}N_{\mathrm{object}})^{d-1}), \tag{11}$$

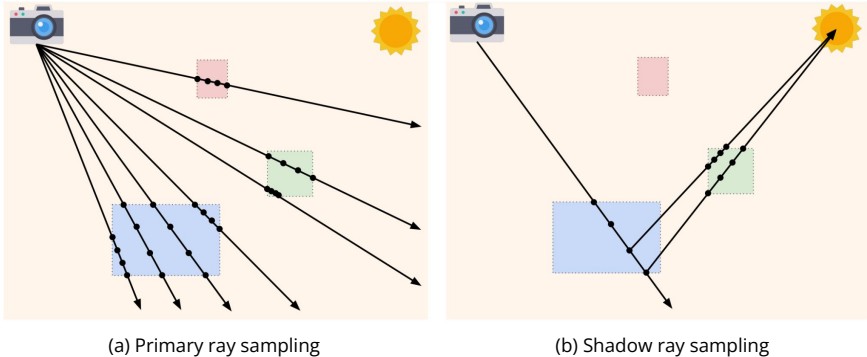

(a) Primary ray sampling      (b) Shadow ray sampling

Figure 3: Sampling procedure. (a) Scene with a camera, light source, and object bounding boxes. Primary rays are sent from the camera into the scene. Rays that do not intersect with objects are pruned. Of the intersecting rays, we sample points within intersecting regions. (b) Shadow rays from each sample are sent to the light source, and samples within intersecting regions are evaluated.

where $N_{\mathrm{sample}}$ denotes the number of samples per ray-object pair, $N_{\mathrm{light}}$ denotes the number of light sources, and $d$ denotes the number of light bounces we model. There are many samples for which we must query OSFs, which can result in significant rendering time. In addition to KiloOSF, we further accelerate scene composition by excluding non-intersecting rays. Specifically, we implement a sampling procedure that precludes the evaluation of rays that do not intersect with objects. Our sampling procedure assumes access to (rough) bounding box dimensions for each object. In practice, such a bounding box can be automatically computed from a trained OSF by extracting the object's bounding volume (using the predicted alpha values). As shown in Fig. 3, for each ray and each object of interest, we intersect the ray with the object's bounding box. If the ray does not intersect with the object's bounding box, then no computation is required for the ray-object pair. If the ray does intersect with the object's bounding box, we adopt the intersection points as the near and far sampling bounds for the ray-object pair. Thus the $N_{\mathrm{pixel}}N_{\mathrm{object}}$ (number of primary rays) and $N_{\mathrm{light}}N_{\mathrm{object}}$ (number of secondary rays) are upper bounds on the number of rays that need to be evaluated. In practice, a single ray often only intersects with at most one object in the scene, which means that the proposed rendering procedure is not significantly more expensive than the single object setting.

Compared with traditional volumetric path tracing methods, our OSF model does not require running path tracing *within* each object (by evaluating itself during secondary ray computations) to simulate intra-object light bounces, because it learns the object-level scattering function that directly predicts the effects after all light bounces (reflections) and occlusions (shadows) within an object have occurred. Therefore, OSFs are faster than alternative methods such as NRFs (Bi et al., 2020a), which relies on simulating intra-object light bounces while querying its learned BRDF model.

## 4 Experiments

We validate our approach by free-viewpoint relighting on both synthetic and real-world objects, including translucent and opaque ones. Furthermore, we also demonstrate that OSFs allow visually plausible scene composition by showcasing a scene with both synthetic and real objects. We leave implementation details and ablation studies in our supplementary material.

### 4.1 Data and baselines

In our experiments, we use six synthetic objects, two captured real translucent objects and five real opaque objects from the Diligent-MV dataset (Li et al., 2020).

**Diligent-MV dataset (Li et al., 2020).** This dataset provides multi-view photometric images of five objects that feature different levels of shininess. For each object, there are 20 views forming a circle around the object. For each view, there are 96 calibrated light sources spatially fixed relative to the camera. We use

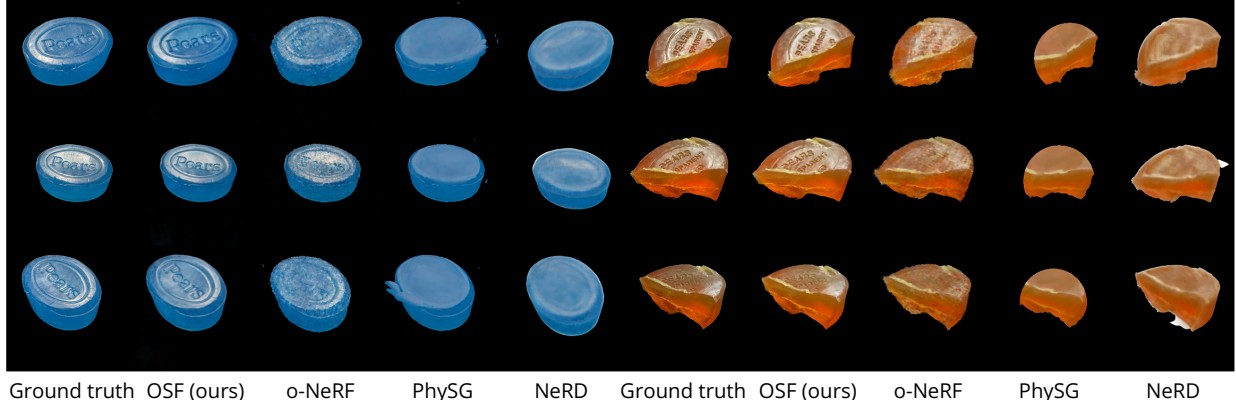

Figure 4: Results on free-viewpoint relighting for real translucent objects captured by a cellphone.

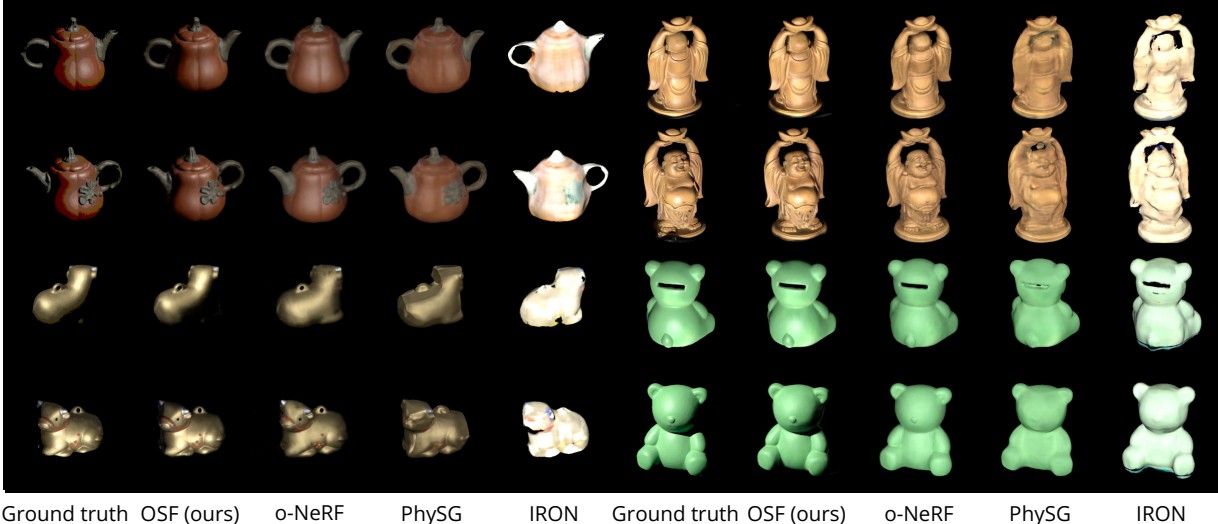

Figure 5: Free-viewpoint relighting for real opaque objects from Diligent-MV (Li et al., 2020).

a front view and a back view (view 10 and view 20) as the testset, and the other 18 views as the training set. Note that the light directions in the training set are completely disjoint from the testset since the light sources move with the camera.

**Real image capture setup.** We use an easily accessible image-capturing setup, where we take photos with two iPhone 12 in a dark room. We use the flash of a cellphone $P_{light}$ as a distant light source. We position $P_{light}$ such that it is far away from the object (the distance to the object is about 10 times compared to the object diameter). We use the other cellphone $P_{image}$ as a camera without using its flash. We capture 20 images from random viewpoints for a single random light direction, and we repeat this for 10 different light directions. This gives us a dataset of 200 images from different viewpoints. We split our dataset such that all 20 images of a randomly chosen light direction are held out for testing and all other images for training. We use a standard Structure from Motion (SfM) method, COLMAP (Schonberger & Frahm, 2016), to solve for camera poses for all images. To calibrate light position, we take a photo using $P_{light}$ for every light position, and solve the camera pose together with other captured images. Photos taken by $P_{light}$ are only used for light direction calibration but not for training. In order for SfM to work well in the dark room, we place highly-textured newspaper pieces surrounding the captured object to provide robust features and use a foreground segmentation network (Qin et al., 2020) to extract clean object images after SfM.

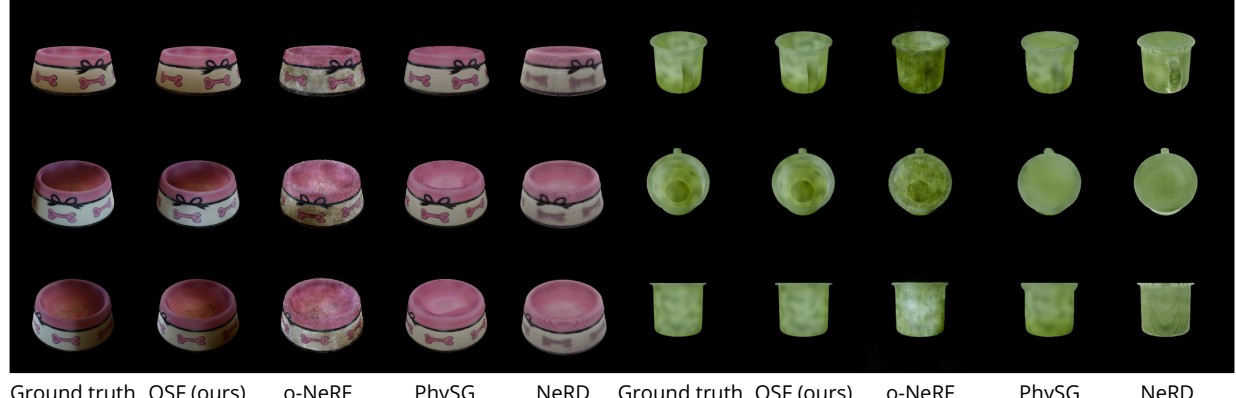

| Ground truth | OSF (ours) | o-NeRF | PhySG | NeRD | Ground truth | OSF (ours) | o-NeRF | PhySG | NeRD |

Figure 6: Free-viewpoint relighting for synthetic translucent objects from ObjectFolder.

|  | PSNR↑ | SSIM↑ | LPIPS↓ |
| --- | --- | --- | --- |
| o-NeRF | 23.69 | 0.811 | 0.137 |
| PhySG | 27.49 | 0.923 | 0.043 |
| NeRD | 24.39 | 0.892 | 0.125 |
| OSF (ours) | **38.99** | **0.982** | **0.006** |

Table 1: Free-viewpoint relighting on synthetic translucent objects.

|  | PSNR↑ | SSIM↑ | LPIPS↓ |
| --- | --- | --- | --- |
| o-NeRF | 17.63 | 0.725 | 0.331 |
| PhySG | 20.09 | 0.823 | 0.171 |
| NeRD | 21.94 | 0.815 | 0.182 |
| OSF (ours) | **27.06** | **0.865** | **0.051** |

Table 2: Free-viewpoint relighting on real translucent soaps.

|  | PSNR↑ | SSIM↑ | LPIPS↓ |
| --- | --- | --- | --- |
| o-NeRF | 31.36 | 0.944 | 0.038 |
| PhySG | 30.09 | 0.944 | 0.046 |
| IRON | 14.80 | 0.907 | 0.081 |
| OSF (ours) | **39.07** | **0.970** | **0.020** |

Table 3: Free-viewpoint relighting on real opaque objects from Diligent-MV (Li et al., 2020).

**Synthetic images.** We use 5 synthetic objects from the ObjectFolder (Gao et al., 2021) dataset, as well as a Stanford bunny. We assign translucent materials to half of them and opaque materials to the other half. For each synthetic object, we render 1,000 images, each of which is under a random light direction and a random viewpoint. We sample cameras uniformly on an upper hemisphere, and light directions uniformly on solid angles. We use 500 images for training and 500 for testing. We generate synthetic images in Blender 3.0, using the Cycles path tracer. We use the Principled BSDF (Burley, 2012) with Christensen-Burley approximation to the physically-based subsurface volume scattering (Christensen, 2015). We also evaluate OSFs on different levels of translucency and on a two-light setting in our supplementary material.

**Baselines.** We consider the following baselines:

*o-NeRF* (Mildenhall et al., 2020): We train a NeRF for each object and refer to it as o-NeRF. Since o-NeRF is agnostic to lighting and cannot do relighting, we show view synthesis results.

*PhySG (Zhang et al., 2021b)*: A recent representative neural inverse rendering method that jointly estimates lighting, materials, and geometry from multi-view images and masks of shiny objects.

*NeRD (Boss et al., 2021a)*: A recent representative neural relightable representation that decomposes the appearance into a neural BRDF field and environment lighting.

*IRON (Zhang et al., 2022)*: A state-of-the-art neural inverse rendering method. Note that IRON assumes collocated lighting for training images and assumes little self-shadow in training images. These assumptions do not hold for most of our data, so we only test IRON on Diligent-MV dataset which has less self-shadows.

We use the following standard metrics in all our comparisons: peak signal-to-noise ratio (PSNR), structural similarity (SSIM) (Wang et al., 2003), and a perceptual metric LPIPS (Zhang et al., 2018).

## 4.2 Free-Viewpoint Relighting

OSFs take light directions as input in addition to the camera viewpoint. Thus, it inherently supports simultaneous relighting and novel view synthesis.

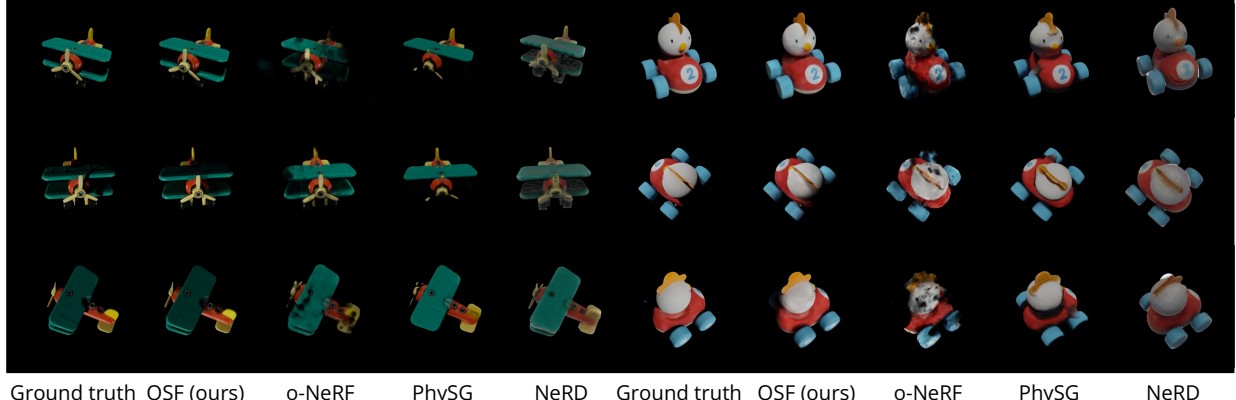

Ground truth   OSF (ours)   o-NeRF   PhySG   NeRD   Ground truth   OSF (ours)   o-NeRF   PhySG   NeRD

Figure 7: Free-viewpoint relighting for synthetic opaque objects from ObjectFolder (Gao et al., 2021).

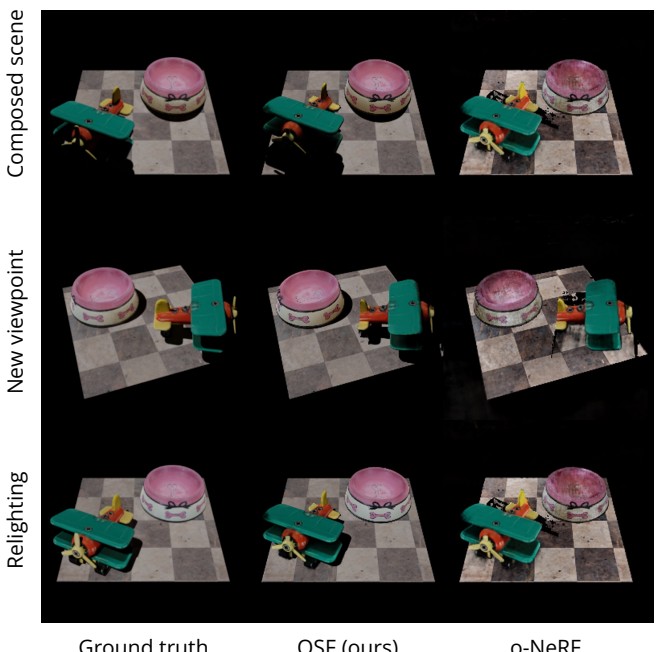

Ground truth   OSF (ours)   o-NeRF

Figure 8: Scene composition for a translucent bowl, an opaque airplane, and an opaque floor.

|  | PSNR↑ | SSIM↑ | LPIPS↓ |
|---|---|---|---|
| o-NeRF | 20.93 | 0.817 | 0.181 |
| PhySG | 24.61 | 0.901 | 0.088 |
| NeRD | 21.12 | 0.866 | 0.106 |
| OSF (ours) | **34.26** | **0.923** | **0.034** |

Table 4: Free-viewpoint relighting on synthetic opaque objects.

|  | PSNR↑ | SSIM↑ | LPIPS↓ |
|---|---|---|---|
| o-NeRF | 17.21 | 0.678 | 0.244 |
| OSF (ours) | **35.57** | **0.927** | **0.031** |

Table 5: Free-viewpoint relighting results for scene composition.

|  | FPS↑ | SSIM↑ | LPIPS↓ |
|---|---|---|---|
| OSF | 0.27 | 0.923 | **0.034** |
| KiloOSF | **16.13** | **0.938** | 0.037 |

Table 6: Evaluation of KiloOSF on relighting synthetic opaque objects.

**Translucent objects.** We show qualitative (Figure 6 and Figure 4) and quantitative (Table 1 and Table 2) comparisons on both synthetic and real translucent objects. Our approach significantly outperforms the baseline methods. It successfully models the material and shape of the objects, and accurately relights them from varied viewpoints compared to the ground truth. In contrast, o-NeRF fails due to its assumption on fixed illumination. Methods that parameterize object materials by BRDFs, such as PhySG and NeRD, fail to produce subsurface scattering effects. Moreover, since they assume opaque surfaces, subsurface scattering effects in the training images seem to be explained by geometry, producing inaccurate geometry reconstruction (e.g., the bowl in Figure 6). Notably, for the real soaps, OSFs can produce correct highlights on the soap surface as well as subsurface scattering effects (e.g., notice how the shading changes non-uniformly along the fracture surface of the orange soap in Figure 4). Furthermore, only OSFs can reproduce the text on the soaps.

**Opaque objects.** We show free-viewpoint relighting results for real objects from Diligent-MV dataset in Figure 5 (we drop NeRD as it has a convergence issue on this dataset despite our best efforts) and

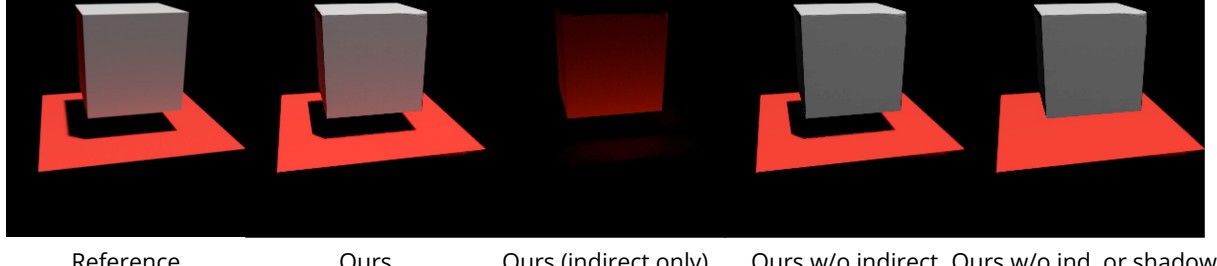

Figure 9: Analysis of light transport effects including shadow and indirect lighting. Note that here we generate the ray-traced reference image with only two light bounces as we only compute two light bounces in our composition. Images are tone-mapped to sRGB space for display.

synthetic opaque objects in Figure 7. We observe that OSFs produce faithful free-viewpoint relighting results, correctly synthesizing hard self-shadows (e.g., the airplane in Figure 7 and the bear in Figure 5) and highlight reflections (e.g., the buddha's belly and the cow in Figure 5). In contrast, baseline methods do not synthesize these effects well. PhySG produces some self-shadows, while it does not reconstruct geometries well on Diligent-MV objects. IRON assumes collocated lighting and it assumes little self-shadows in training images. Thus it does not reconstruct or relight the objects correctly. We show quantitative results in Table 4 and Table 3, where OSFs significantly outperform all baselines.

### 4.3 Scene Composition

Since OSFs are object-centric representations that model light transport for an object, they allow trained models to be readily composed into new scenes. To showcase neural scene composition, we use a translucent and an opaque object with randomized poses, lighting directions, and viewpoints. We show relighting and view synthesis of this example scene in Figure 8. We compare with the *o-NeRF* baseline. Qualitatively, while *o-NeRF* produces incorrect shadows and artifacts, our method generates more realistic compositions that better match the ray-traced groundtruth images. We also show quantitative results in Table 5, where we observe that OSFs significantly outperform *o-NeRF*.

**Light transport analysis.** To analyze the light transport effects including shadows and indirect lighting in our composition method, we showcase a simple scene consisting of a grey cube and a red floor in Figure 9, where we show our scene composition and variants ablating shadows and indirect lighting. From Figure 9 we observe increased realism when we add shadow and indirect lighting. Specifically, we see a color bleeding effect from the red floor onto the cube by the indirect lighting. With these light transport effects, it correctly synthesizes the scene up to two light bounces (i.e., direct lighting and one-bounce indirect lighting). While here we showcase a simplistic scene and compute one bounce for indirect lighting due to computational constraints, our method may benefit from future advances in efficient neural rendering to scale to more complex scenes and more light bounces.

### 4.4 Evaluation on KiloOSF

We introduce KiloOSF for accelerating OSF rendering. Thus, in this section, we evaluate KiloOSF in terms of rendering speed and relighting performance. To this end, we test KiloOSF on all synthetic translucent objects. We show the rendering speed (frames per second) and the rendering quality in Table 6. KiloOSF achieves $60\times$ (16.13 vs. 0.27 FPS) speed up while maintaining comparable relighting performances, suggesting the benefits brought by the KiloOSF design.

## 5 Conclusion

We presented a novel, learning-based object representation—Object-Centric Neural Scattering Functions (OSFs). An OSF models a cumulative radiance transfer function, allowing both free-viewpoint relighting

and scene composition. Moreover, OSFs can model objects of complex shape and materials, and can relight both opaque and translucent objects. Our work offers a new promising neural graphics method for modeling real-world scenes.

**Limitations.** The major limitation of our model comes from the assumption of unobstructed distant lighting, which leads to biased estimation in, e.g., computing inter-reflection, especially considering shiny objects. In real capture, distant light can be approximated by holding a flashlight far away from small objects, but for big objects this is difficult. We envision wider applications in future research that relaxes the assumption.

**Acknowledgments.** This work is in part supported by Google, NSF RI #2211258, ONR MURI N00014-22-1-2740, Stanford Institute for Human-Centered Artificial Intelligence (HAI), Amazon, Bosch, Ford, and Qualcomm.

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
