# OpenReview forum: "Learning Object-Centric Neural Scattering Functions for Free-viewpoint Relighting and Scene Composition"
_TMLR — Accepted by TMLR_

### Review · Reviewer_XktD · 2022-12-23

**Summary Of Contributions:**

This work proposes Object-Centric Neural Scattering Functions (OSFs), a novel formulation to model photorealistic object appearance from collections of 2D images with neural implicit models. By taking an incoming light direction and subsurface scattering effects into consideration, OSFs support modeling both opaque and translucent objects, and can relight object appearance using a novel light source. In addition, multiple learned OSFs can be composed together and render a complex scene of multiple objects. Photorealistic scenes are visualized and quantitative results are included to demonstrate the efficacy of OSFs in modeling object appearance and composing new scenes.

**Audience:**

Yes

**Broader Impact Concerns:**

No concerns.

**Claims And Evidence:**

Yes

**Requested Changes:**

In addition to the weaknesses described in the section above, the following changes can be made to strengthen this work:

- The baseline o-NeRF does not take the light source into account. It would be better to explain how to relight with this method in the experiments.

- It would be more convincing to generate more scene dataset with more challenging visual effects and evaluate them.

- In Table 5, the PSNR of two methods are abnormally small compared to other tables (the MSE before calculating PSNR will be ~ 0.99). I wonder if there is a mistake in this table.


**Strengths And Weaknesses:**

Strengths:

- This work studies a novel, challenging setting beyond traditional NeRF that requires the model to support relighting. By considering the direction of the light source, OFS allows synthesis with both novel viewpoints and novel light sources.

- The proposed OSF is a novel method that natively supports light transport effects like shadows.

- This work shows successful modeling of translucent objects, which is a major challenge in previous work like NeRF and other neural inverse rendering methods.

- By combining with the idea of KiloNeRF, an efficient version of OSF can be implemented to accelerate rendering, achieving 60x speed up with only slight loss of rendering quality.

Weaknesses:

- One limitation of the proposed setting which OSF studies is that it only allows **single distant light source**. In practice, there can be multiple light sources contributing to a more realistic scene, and they may be ambient or point light sources. Currently, OSF cannot model such light sources.

- Another challenge lies in the real-world data collection process. To generate a training dataset for OSF, one has to capture images of objects in a dark room, and also calibrate the light position with SfM methods. This data collection pipeline is significantly more complicated than standard NeRF and requires more expertise. Synthetic data can be generated as easily as traditional NeRF though.

- In Table 1, OSF performs slightly worse than the baseline PhySG in terms of the SSIM metric, which contradicts the claim (in the text) that “our approach significantly outperforms the baseline methods”. Though, OSF indeed outperforms other methods with a higher PSNR. It would be better if authors explain why the trend on SSIM is different from other metrics, or check if the computation of SSIM is inconsistent.

- The dataset for evaluation is too simple. It does not include challenging visual effects like mirror reflections or light spots, even few view-dependent effects. I wonder if OSFs can still perform well with these effects.

---

### Review · Reviewer_QtbL · 2023-01-11

**Summary Of Contributions:**

This work presents a neural-field approach to capturing relightable 3D representations of objects. The proposed approach uses an MLP to map position, (incident) light direction, and camera/view direction to a radiance and density; the radiance measures the total (cumulative) radiance emitted at the relevant point/angle due to a distant light-source in the given direction, including any light transported to the observed point via subsurface scattering and/or intra-object reflections. This 'pre-integration' of radiance contrasts with a traditional (or indeed neural) BSDF, and allows efficient rendering of the object (including subsurface effects) without integrating over light-paths. The proposed representation also differs from NeRF in that it includes a dependence on light direction, hence supports relighting. Additionally, the paper discusses an approximate method for composing multiple pre-captured objects together in a scene, accounting for (limited) global/indirect illumination effects. The method is demonstrated on several synthetic and real captures, where it out-performs a number of baselines quantitatively and qualitatively.

**Audience:**

Yes

**Claims And Evidence:**

No

**Requested Changes:**

### Critical

- please provide higher-resolution / less-compressed versions of the renderings in supplementary, since it is currently difficult to make detailed judgements of quality/fidelity.

- discuss and/or evaluate against the references mentioned above

- evaluate on a standard dataset, e.g. Diligent-MV

- provide some convincing examples of accurate compositions including indirect illumination

- fix tab. 5

### Non-critical

- evaluate on more objects, or at least say how/why the specific synthetic objects for evaluation were chosen

- I assume objects were pre-segmented / masked to remove background; however this doesn't seem to be stated. Please clarify somewhere

- sec 3.6 seems to apply exclusively to the compositional setting discussed in sec 3.5 -- it would therefore be clearer if 3.6 were a subsubsection of 3.5. Alternatively 3.5 & 3.6 could be moved to a separate top-level section.

- tab. 3-5 are very squeezed, above fig. 7 which has a lot of whitespace. Perhaps this page could be arranged more efficiently

**Strengths And Weaknesses:**

The proposed framework is novel, and addresses an interesting problem. The different components of the system are reasonably motivated, and explained clearly.

Pre-integrating the radiance is not a new idea (as the paper notes); however I am not aware of any work that applies it in the exact setting here. It enables much faster rendering than "more local" representations such as (neural or traditional) BSDFs, which require tracing many light-paths when rendering.

The text is generally balanced, and explicitly acknowledges various limitations and their impact in the main text (e.g. strong assumptions on the incident lighting). I particularly appreciate this given the tendency of many works to 'bury' such information at the end!

The work of Zheng et al. (2021) is mentioned in sec. 2, but it is not very clear why exactly the proposed work is better than theirs (which is quite similar in task/setting). Please add either a theoretical justification (e.g. inability of theirs to model certain effects) or an empirical one (e.g. proposed is faster / more accurate).

"IRON: Inverse Rendering by Optimizing Neural SDFs and Materials from Photometric Images" [Zhang, CVPR '22] is very relevant, and I think SOTA for the multi-view multi-light setting considered in the paper (but it doesn't support subsurface scattering). It should at least be cited, and maybe compared against.

The quantitative evaluation appears to use appropriate protocols and metrics. However, only five synthetic objects are used, and it's not clear how these were selected (the underlying dataset is larger). Similarly, there are very few qualitative examples shown -- particularly for scene composition. Those which are present, demonstrate good performance, but it is difficult for the reader to get an idea of how well the method works in general on diverse objects/scenes.

For the real data captured with two phones, the train/evaluation split is done by holding out all viewpoints with particular light directions. This is clearly sufficient when measuring performance on relighting. However, when considering novel view synthesis, it is important that the camera poses of the held-out images are not too close to those from the training split -- there is no indication of whether this is the case, save for the statement that the camera was moved randomly.

A benchmark dataset such as Diligent-MV (typically used for multi-view photometric stereo) should be used to give a more extensive (and standardised) evaluation of performance on real-world images, at least in the case of opaque surfaces.

It is mentioned in sec 3.5 that when composing multiple objects in close proximity, indirect illumination effects are not accurately captured by assuming infinite distance sources of radiance. This would be particularly true if the objects were more glossy, and hence sharper reflections of nearby objects should be seen. This seems like a significant limitation of the proposed approach. I'd be happier if the method were claimed simply to model individual objects under certain lighting conditions; however the fact that the composability is called out as an explicit benefit makes this more problematic (since the chosen representation is not so great for this).

The qualitative results on composition in fig. 8 are not particularly impressive, in terms of level of detail and realism of the composition (however, this is difficult to evaluate rigorously given the highly-compressed / low-resolution images). More problematically, the "indirect only" results appear very wrong. For example on the top surface of the highest object, there should be zero indirect contribution; similarly on all surfaces that are entirely (self-)occluded from other objects, there should also be zero indirect illumination. However, the figure seems to show a fairly uniform indirect component over most surfaces, regardless of orientation. It would be much more convincing if a scene were shown where some stronger indirect illumination effects should be seen -- c.f. the "Cornell box" with some strongly-colored object adjacent to a white one, showing clear scatter onto the latter.

The PSNR values in sec. 4.4 / tab. 5 seem to be wrong -- they are not in the correct range (by a large factor).

A small weakness of the overall approach is that it assumes incident light directions are known. In the real-world experiments authors solve this by using a phone for the light, and taking a photo; however in practical capture scenarios the light-source may not have a camera attached. It would therefore be valuable to measure how sensitive the method is to errors in estimation of the lighting angles (e.g. by adding increasing levels of gaussian noise to the angle, and measuring how PSNR is affected). Moreover, the lights are assumed to be infinitely-distant (i.e. purely directional) sources. Again this is not satisfied in real-world imaging conditions, and it would be valuable to determine how sensitive the model is to this (rather than merely observing that it works 'well enough' with a phone flash at a certain distance).

The paper is well-written and pleasant to read throughout. The structure is fine, and figures/etc. are legible and appropriate.

---

### Review · Reviewer_J5zP · 2023-01-29

**Summary Of Contributions:**

This work presents a new neural object representation that models translucency and allows for novel view synthesis, relighting, object composition. Modeling translucency is the main distinction, which is possible due to a cumulative radiance transfer function formulation to model the appearance of objects. This formulation uses the neural network itself to implicitly learn the complex subsurface scattering properties of an object by modeling output radiance at a spatial location as a function of the radiance and direction of a distant light source and the outgoing light direction. This makes it possible to learn this representation from calibrated 2D images and light sources with known positions, without additional information about the shape of the object or its material properties.

Qualitative and quantitative evaluation is done on 5 synthetic objects from ObjectFolder and 2 real objects. The proposed method outperforms the standard NeRF and additional baseline methods PhySG and NeRD.

**Audience:**

Yes

**Broader Impact Concerns:**

There are no broader impact concerns.

**Claims And Evidence:**

Yes

**Requested Changes:**

Critical changes:
- Real world evaluation on a 2-3 more objects with more complex geometry. This is important for understanding the limitations of the proposed method.
- A detailed description of the synthetic data generation process (could go in the supplement) and an explanation behind the design decisions for this process (e.g. how were objects chosen, how were camera and lighting positions chosen, how were the translucent object materials implemented)

Changes that would strengthen the work:
- Carefully using the synthetic data to understand the connection between object geometry, translucency and how well the proposed approach can model them.


**Strengths And Weaknesses:**

Strengths:
- The paper is overall well written and straightforward to follow The draft is in good shape and the supplementary video is an excellent summary of the proposed work.
- The approach is well motivated. Explicitly modeling subsurface scattering is difficult and existing neural field formulations do not propose ways of handling this, making them unsuitable for translucent objects. Using a neural network to model the complete process of how an object responds to a known light source makes sense.
- In addition to proposing how to estimate cumulative radiance transfer with a neural network, the method contains additional improvements: modeling indirect lighting, an efficient ray sampling procedure, and a KiloNeRF-based extension that allows for fast inference.

Weaknesses:
- The proposed approach requires both images with known viewpoints and known light source directions. In the context of prior neural field works this limitation is OK given how difficult it is to model translucency. This requirement requires a more complex capture process and limits the method to indoor scenes of objects.
- The real world evaluation is limited: this is done only on two simple, mostly convex candy-like objects. While it is clear that the proposed approach outperforms prior work, real world evaluation on objects with more complex geometry would help us better understand the limitations of the current approach.
- Synthetic evaluation can be improved: how were the objects used for evaluation chosen, and how exactly were the translucent objects made to be translucent? What specific material parameters were chosen and why? The power of synthetic data is the fine grained control over aspects of the visual world that affect how a model learns to represent it. It would be very useful to vary the translucency parameters of objects in the synthetic domain and understand how well the proposed approach can model them and potentially identify some failure modes. This would be extremely useful for future work.

---

### Review · Reviewer_sjM8 · 2023-01-30

**Summary Of Contributions:**

The paper tackles the challenge of scene reconstruction from 2D image inputs. Following the success of NerF-based models, the paper suggests extending the underlying rendering procedure of NerF to take into account also the rendering of translucent objects. To do so, subsurface scattering effects are modeled by predicting the cumulative radiance in a specific view as an integration over all possible directions.
The method is evaluated on both synthetic and real captured scenes.


**Audience:**

Yes

**Claims And Evidence:**

Yes

**Requested Changes:**

Please see the above weakness, they list some comments to be considered in the next revision.

**Strengths And Weaknesses:**

Paper Strengths

I appreciate the effort to tackle the challenging problem of translucent object simulation.

The introduction section provides a clear motivation for the method.

An efficient computational model based on KiloNerf is incorporated as well.

Paper Weaknesses

There should be an effort to make the formulation in the method section clearer.
Some examples:
What does {} stands for in equation (2)?

In equation (1), r is a function of t, but t is also the integrand variable. So, what is L(r) in the LHS?

In eq (8), what are the loss parameters to be minimized?


More Evaluation is needed.

The evaluation could be more extensive. It would be beneficial to evaluate the method on popular benchmarks such as the DTU MVS repository (Jensen et al. ).
Moreover, it seems that the baselines chosen cannot overfit specific details in the input images (iike the captions in figure 5).  It is unclear to me why NerF would fail to overfit these details, as I can imagine it can “encode” these details in the color network for specific views and only fail in generalization to novel views. It would be beneficial to provide rendering results for training images as well.

Missing discussion about the method

It seems that the main idea suggested is to replace the computationally intractable integral in equation (6) by evaluating it on one predefined light view. What are some immediate limitations that such an approximation yields? It would be beneficial to discuss and provide some examples.

---

### Decision · Action_Editors · 2023-03-28

**Recommendation:** Accept as is

**Comment:**

3 reviewers who have submitted their final recommendations have all agreed on accepting the paper. The 4th reviewer who has not submitted the final recommendation had similar initial concerns with the other 3. Based on the rebuttal and discussion, AE believes those concerns were addressed as well. Hence, AE recommends acceptance of this paper.

**Audience:**

Yes. NERF-based representations are important use cases of using machine learning approaches to solve an optimization problem. The approach this paper proposes adds value by addressing translucent object reconstruction and includes several additional improvements.

**Claims And Evidence:**

This paper tackles the difficult problem of translucent object reconstruction. After initial reviews and rebuttal, 3 of the reviewers are in favor of acceptance and the 4th reviewer failed to submit their final rating. The initial concerns about evaluation have been addressed by the authors via additional experiments on real data and additional ablation studies. The reviewers find the paper well-written and well-motivated. Besides, In addition to proposing how to estimate cumulative radiance transfer with a neural network, the method contains additional improvements: modeling indirect lighting, an efficient ray sampling procedure, and a KiloNeRF-based extension that allows for fast inference.

Based on the above, the AE recommends acceptance of the paper.